# Gravity-Driven Separation of Oil/Water Mixture by Porous Ceramic Membranes with Desired Surface Wettability

**DOI:** 10.3390/ma14020457

**Published:** 2021-01-19

**Authors:** Chunlei Ren, Wufeng Chen, Chusheng Chen, Louis Winnubst, Lifeng Yan

**Affiliations:** 1School of Chemistry and Materials Science, University of Science and Technology of China, Hefei 230026, China; chorcl@mail.ustc.edu.cn (C.R.); cwf@mail.ustc.edu.cn (W.C.); 2Inorganic Membranes, MESA+ Institute for Nanotechnology, University of Twente, 7500 AE Enschede, The Netherlands; a.j.a.winnubst@utwente.nl

**Keywords:** oil/water separation, porous alumina, FAS grafting, hydrophilic/underwater oleophobic, hydrophobic/oleophilic

## Abstract

Porous Al_2_O_3_ membranes were prepared through a phase-inversion tape casting/sintering method. The alumina membranes were embedded with finger-like pores perpendicular to the membrane surface. Bare alumina membranes are naturally hydrophilic and underwater oleophobic, while fluoroalkylsilane (FAS)-grafted membranes are hydrophobic and oleophilic. The coupling of FAS molecules on alumina surfaces was confirmed by Thermogravimetric Analysis and X-ray Photoelectron Spectroscopy measurements. The hydrophobic membranes exhibited desired thermal stability and were super durable when exposed to air. Both membranes can be used for gravity-driven oil/water separation, which is highly cost-effective. The as-calculated separation efficiency (*R*) was above 99% for the FAS-grafted alumina membrane. Due to the excellent oil/water separation performance and good chemical stability, the porous ceramic membranes display potential for practical applications.

## 1. Introduction

Separation of oil and water mixture is becoming a worldwide challenge due to the increasing amount of industrial oil-containing wastewaters and oil spill accidents [1]. Currently, the separation of oil/water mixture is mainly achieved by making use of the difference in density between oil and water. For example, separation can be achieved through gravity settling or centrifugation, but these methods suffer from large area occupation or high energy consumption. Membrane separation has been accepted as a clean and effective process for the treatment of wastewater in the past decades. In principle, the oil/water separation can be attained by making use of porous materials with the desired surface wettability. For example, Yang et al. prepared robust hydrophobic cotton fabrics by anchoring polyhedral oligomeric silsesquioxane onto the membrane surface [2]. Porous PVDF (polyvinylidene fluoride)/graphene and ZIF (Zeolitic Imidazolate Framework) modified composite membranes are also applied for oil removal from water [3,4]. These hydrophobic and oleophilic porous materials allow the oil to pass through while water is retained, resulting in the separation of oil dispersed in water [5,6,7]. Likewise, water can be separated from a mixture using hydrophilic and oleophobic porous materials. Gao et al. prepared a nitrocellulose membrane with underwater superoleophobicity for highly efficient oil/water separation [8]. Recently, a couple of modified meshes have been made for energy-efficient oil/water separation [9,10,11,12]. These methods are easy to scale up and are potentially cost-effective. In comparison with polymers and metals, ceramics exhibit better chemical and thermal stability, robust mechanical durability, and corrosion resistance. These features endow the ceramic membranes with more advantages in a harsh environment.

The phase inversion method was first described by Loeb and Souririjan [13] for the preparation of asymmetric polymeric membranes. The exchange between solvent (in polymer precursor) and non-solvent (coagulation bath) leads to precipitation of the polymer phase and the formation of finger-like voids and a sponge-like structure. The phase inversion method has been adopted for ceramic membrane preparation by adding the desired amount of ceramic powder to the polymer solution [14]. Ceramic particles are immobilized once precipitation takes place and the macrostructure of the precursor can be retained during heat treatment. The membrane macrostructures can be largely determined by adjusting the various parameters of the phase inversion process [15]. Due to the unique pore structures created during the phase inversion process, the ceramic membranes exhibit small mass transfer resistance and hence display great potential for highly efficient separation processes. Recently, phase inversion combined with the tap casting/extrusion and sintering method was employed to prepare planar/tubular ceramic membranes for water desalination [16], anode supported solid oxide fuel cell [17], and oil/water separation [18].

Ceramic membranes are usually made of metal oxides, thus are naturally hydrophilic due to the surface hydroxyl (–OH) groups. Consequently, water is readily absorbed on the membrane surface and permeates into the ceramic pore matrix, while oil droplets can be repelled by the surface hydroxyls and the absorbed water thin film. This character allows these ceramic membranes to be suitable for oil/water separation where water content is relatively low. On the other hand, the hydrophilicity of the ceramic membranes has traditionally made them unsuitable for the separation process where a small amount of oils needs to be removed from the water. Various materials have been used to modify ceramic membranes to make them hydrophobic and oleophilic [19,20,21]. Among these materials, fluoroalkylsilane (FAS) is mostly used as a surface coupling agent for ceramics. The functionalization by fluorinated alkyl moieties such as –CF_2_–, –CF_2_H, and –CF_3_ are quite effective for lowering surface free energy [22,23]. The grafting process can be performed by a reaction between the –OH surface groups of the ceramics and ethoxy groups (–O–Et) presented in organosilane compounds [24]. However, the coupling strength of FAS on metal oxides, or the durability of membrane hydrophobicity, is still unclear and needs to be further studied before the membranes are applied in the oil/water separation industry.

The oil state in the presence of oilfield sewage can be generally classified into emulsified oil (d_p_ < 10 μm), dispersed oil (10 μm < d_p_ < 100 μm), and floating oil (d_p_ > 100 μm), according to the oil droplet diameter (d_p_) [25]. The oil emulsion is stable and can be hardly treated by the traditional mechanical separation methods, where nanofiltration and microfiltration membranes are the two main improved technologies for demulsification [26]. Floating oil with the largest droplet size is relatively easier to separate from water, and are usually treated by traditional methods. Few membrane processes have been developed to treat floating oil/water separation. It is worth noting that floating oil accounts for a large part of the total oil in typical oilfield wastewater. As a result, a desired membrane separation strategy with low cost and reliable materials can be used as a green alternative to the existing commercial separation technologies.

Here, we report a porous alumina membrane, containing straight finger-like pores ranging from 10 to 30 μm in diameter, which are suitable for the fast separation of floating oil/water mixture. The bare alumina membrane is hydrophilic in nature and can be used to remove water from its mixture with oil. After grafting with fluoroalkylsilane (FAS), the porous alumina membrane becomes hydrophobic, thus can be used to remove oil from its mixture with water. The FAS coupling property and durability on the alumina membranes have been systematically studied. Importantly, the porous alumina used in this study contains large finger-like pores along the thickness direction, typical for pore structures formed using the phase-inversion method. Due to the extremely small resistance to mass transfer, the porous ceramics can use gravity solely as the driving force for the separation of oil and water.

## 2. Methods

### 2.1. Preparation of Planar Alumina Membranes

The phase-inversion tape casting method was used to fabricate planar alumina membranes [27]. Polyethersulfone (PES, polymer binder) (6.2 wt.%) (Gafone 3000) (Solvay Advanced Polymers, Shanghai, China) and polyvinylpyrrolidone (PVP, additive) (0.9 wt.%) (CP, Sinopharm Chemical Reagent Co., Shanghai, China) were first dissolved in N-methyl-2-pyrrolidone (NMP, solvent) (31.0 wt.%) (CP, Sinopharm Chemical Reagent Co., Shanghai, China). After the polymer solution was formed, α-Al_2_O_3_ powder (D_50_ = 0.72 μm) (61.9 wt.%) (Zhenghai Ceramic Co., Shanghai, China) was added into the solution and stirred for two days. The as-prepared suspension was degassed and then cast on a Mylar sheet with a doctor blade with a gap height of 1.0 mm and motion speed of 20 cm/min (Figure 1a). The casted slurry was solidified by immersion in water for 12 h at room temperature. The tape was then dried at room temperature and cut into round pieces with a diameter of 40 mm (Figure 1b,c), then heated at a rate of 2 °C/min to 800 °C and kept at this temperature for 4 h to remove the polymer binder, followed by heating to 1500 °C at a rate of 2 °C/min and being held for 10 h. The samples were cooled down to room temperature at a rate of 2 °C/min. The whole sintering procedure was processed in the air. After sintering, both sides of the wafer were mechanically polished to a controlled depth using an automatic ceramic polisher (KEMET15, Guangzhan Precision Machinery, Suzhou, China).

### 2.2. Surface Grafting of Alumina Membranes with Fluoroalkylsilane (FAS)

Bare alumina membranes were immersed into a 2 wt.% FAS (1H, 1H, 2H, 2H-perfluorooctyltriethoxysilane) ethanol solution at room temperature for a total grafting time of 72 h that was divided into three successive periods. After each of these periods, the membranes were dried in an oven at 100 °C for 6 h.

### 2.3. Instruments and Characterization

SEM(scanning electron microscope) images of the membranes were obtained using a field-emission scanning electron microscope (JSM-6700F, Tokyo, Japan). The water contact angle of each sample was determined using a contact angle meter (JGW-360A, Chenghui, China) at ambient temperature. The volume of a deionized water droplet was about 5 μL, and the average value of five measurements was adopted as the contact angle. Octane and water were dyed with methylene blue and oil red O, respectively. Diffused Reflectance Infrared Fourier Transform Spectroscopy (DRIFTS) of the alumina powders before and after grating with FAS were performed on a 6700 FTIR spectrometer (Nicolet, Shanghai, China). FAS grafted α-alumina powders with different grafting times were analyzed using a thermal gravimetric analyzer (TGA, SDT Q600, New Castle, DE, USA). The surface composition of the samples was determined by XPS (ESCALAB-250, Carlsbad, CA, USA). The membrane performances including octane/water permeation and oil/water separation were tested using self-made devices, The oil concentration of the collected water in the permeate was measured by gas chromatography (GC, Agilent 6890N, Santa Clara, CA, USA), where CCl_4_ was used to extract oil from water. The peak area around 4.1 min is proportional to the amount of octane in CCl_4_. By comparing the peak area with a standard curve, the oil content in the permeate was obtained.

## 3. Results and Discussion

Porous alumina membranes used for this study were fabricated through the phase-inversion tape casting method as illustrated in Figure 1. The dried green tape was cut into the desired shape and size (Figure 1b,c), followed by sintering and mechanical polishing to controlled thickness. The alumina membrane has a typical structure consisting of a top side skin layer, finger-like pore layer, and bottom side sponge layer (Figure 2a). After polishing, both the skin layer and sponge layer were removed, resulting in well arrayed open pores from top to bottom along the thickness direction (Figure 2b). The resultant alumina membrane showed a gradient distribution of pore size along the thickness direction: at the top surface 15~30 μm (Figure 2c), and at the bottom side 70~100 μm (Figure 2d). Due to the large pore size and small tortuosity, the as-prepared ceramic membrane is expected to present a small resistance to mass transfer.

The surface wettability of the as-prepared porous alumina was characterized. As shown in Figure 3a, both water and oil droplets sunk into the pores, revealing that the alumina disk was hydrophilic and oleophilic in the air. The alumina disk was also found to show underwater oleophobicity with an oil contact angle of ~130° (inset, Figure 3a). The occurrence of underwater oleophobicity can be explained as follows [28]. Due to the presence of hydroxyl groups at the ceramic surface, water molecules are adsorbed to their balanced state. Consequently, the ceramic disk is covered by a water film. When the ceramic disk is then brought into contact with oil droplets, an oil/water/solid composite interface is created. The wettability of the ceramic membrane can also be changed by grafting with FAS. As shown in Figure 3b, the oil droplet was quickly absorbed while the water droplet stayed on the surface with a contact angle of 145–148°, indicating that the FAS-grafted alumina membrane was approximately superhydrophobic (a surface with apparent water contact angle larger than 150° [29,30,31]). The superhydrophobicity of the FAS grafted membrane can be explained by the well-established Wenzel and Cassie Model, which describes the relationship between apparent contact angle (*θ_w_*) and Young contact angle (*θ*_0_) [32,33,34]. Increasing the surface roughness and/or lowering the surface free energy will greatly increase the apparent water contact angle on a hydrophobic membrane (Figure 4a).

The chemical composition of the FAS on the alumina surface was investigated by Diffused Reflectance Infrared Fourier Transform Spectroscopy (Figure 5). For un-grafted alumina powder, broad O–H peaks are observed at about 3400 cm^−1^, indicating the hydrophilic nature of alumina. The intensity of O–H peaks decreased for FAS grafted surfaces, confirming an increase in the hydrophobicity. The strong absorption peak at around 1050 cm^−1^ corresponded to the vibration modes of Al_2_O_3_ [35]. For the FAS grafted alumina, peaks observed between 1200 to 1500 cm^−1^ were due to stretching and bending of C–F bonds, while bands between 2700 to 3000 cm^−1^ originated from the C–H stretching vibrations in FAS [36].

The thermal stability of the FAS coating on alumina surfaces was evaluated. Figure 6 shows the weight loss processes for the FAS-grafted alumina powders with different grafting times. The weight loss of the samples from 100 to 250 °C can be attributed to the desorption of surface absorbed water and the amount of the FAS on alumina surfaces was proportional to the weight loss ranges from 250 °C to 600 °C. It is reasonable that the sample grafted with FAS for 72 h showed the smallest weight loss of water and the largest weight loss of FAS. Two stages of weight loss were found for the FAS grafted samples (G3), one from 250 °C to 350 °C and the other from 350 °C to 600 °C. Considering that the boiling temperature of FAS is 270 °C, which is a little bit higher than the initial weight loss temperature (250 °C), the first stage of weight loss might be related to the evaporation of physically absorbed FAS molecules. In comparison, chemically bonded FAS molecules on an alumina surface through Si–O–Al bonds are more stable and would decompose at elevated temperatures (e.g., above 350 °C). Meanwhile, the self-condensation between FAS molecules through Si–O–Si bonds could also increase the gasification temperature of coating layers [37].

The durability of the FAS-grafted alumina membrane was tested by exposing the membrane to air, water, and ethanol, respectively, for a long period (two months). The water contact angle was periodically measured as a function of the elapsed time, as shown in Figure 7. In all three exposure tests, the FAS grafted membrane showed good hydrophobicity during the first couple of days and began to decrease significantly thereafter. The contact angel decrease was caused by the accumulated water adsorption on the membrane surface and the FAS dissolution in solvents. The membrane maintained good hydrophobicity with a water contact angle of 120° after exposure to air for 60 days. In comparison, the membrane became hydrophilic after 40 days of exposure to water or 30 days of exposure to ethanol. Given that the FAS coating on the alumina surface can be periodically re-generated through the simple modification process, our FAS-grafted membrane is acceptable for practical applications.

Ultrasonic cleaning with ethanol was performed for the FAS-grafted membrane to study the membrane cleaning tolerance. The membrane was immersed in ethanol, ultrasonically cleaned for 5 min, and dried at 100 °C in one cycle. The cleaning cycle was performed three times and XPS analysis was carried out after each cleaning cycle. Figure 8 shows the F 1s profiles on the FAS grafted membrane before and after cleaning. The F 1s peak intensity is proportional to the FAS amount on the membrane surface. It was found that the FAS amount was significantly reduced after the first and second cleaning and remained stable after the third, indicating that most of the FAS molecules were weakly coupled on the membrane surface, and only a small portion of the FAS molecules were chemically bonded to the alumina surface. As a result, the FAS layer needs to be re-generated regularly to maintain the desired hydrophobicity in practical applications.

Due to the unique pore structure in combination with desired surface wettability, both porous alumina membranes were used for oil/water separation. The permeation experiment was conducted using the setup schematically shown in Figure 9, and the driving force for permeation was provided solely by gravity without any external force, as expressed by:(1)ΔP=ρgh
where *ρ* is the density of the liquid; *g* is the acceleration of gravity; and *h* is the height of the liquid column above the membrane. Pure alumina membranes or FAS grafted membranes were fixed at the bottom of the glass tubes with lengths of 13 cm, 23 cm, 33 cm, and 38 cm. During the liquid permeation experiments, the liquid level was kept unchanged by refilling the tube with the test liquid, in order to keep a fixed driving force. It was found that the water permeability for the un-grafted bare alumina membrane was as high as 14.7 L m^−2^ s^−1^ kPa^−1^. Due to its underwater oleophobicity, octane did not permeate through the membrane until the octane height reached ~16.0 cm, corresponding to an intrusion pressure (*P_exp_*) of ~1.1 kPa (Figure 10a), as calculated using Equation (1). For the FAS-grafted membrane, a permeability as high as 14.1 L m^−2^ s^−1^ kPa^−1^ for octane was measured (Figure 10b). Due to the hydrophobicity for the FAS-grated membrane, the water did not permeate at the water height below ~32.5 cm, corresponding to a water intrusion pressure of ~3.2 kPa. Note that the as-measured intrusion pressure agreed well with the theoretical value (3.1 kPa) calculated from the following equation [28,38]:(2)$$P_{theor}=2\gamma_{LV}cos\theta_{0}/d$$
where $\gamma_{LV}$ is the water/vapor interfacial tension (7.28 × 10^−2^ Nm^−1^, 20 °C); *θ*_0_ is the water contact angle on a flat surface (~131°, see Figure 4b); and *d* is the maximum diameter of the top surface pore (~30 μm).

The separation performance of the porous alumina membranes was tested using the setup shown in Figure 10. In one experiment, the bare alumina membrane was fixed at the bottom of a glass tube with a height of 13 cm (Figure 11a) and a mixture of octane and water (30/70 *v*/*v*, totally 170 mL) was poured into the tube. Water permeated quickly through the membrane (Figure 11b). Meanwhile, the octane was repelled and retained in the feed due to the underwater oleophobicity of the membrane. The whole separation process was completed within 90 s. No visible oil existed in the permeated water and the amount of oil in it was further measured by gas chromatography. The separation efficiency was calculated by the oil rejection coefficient *R* according to:$$R\left(\%\right)=\left(1-\frac{C_{p}}{C_{0}}\right)\times 100$$
where *C*_0_ and *C_p_* are the oil concentration (g/L) in the oil/water feed mixture and the permeate, respectively. The as-calculated separation efficiency (*R*) was ~96.0%, indicating that the bare membrane could remove most of the oil from the mixture.

Another separation experiment was conducted with a FAS-grafted membrane using a mixture of octane and water (70/30 *v*/*v*, totally 170 mL). As shown in Figure 11c, the mixture was slowly poured into the tube. Oil permeated quickly through the membrane, while water was repelled above the membrane and flew into a beaker. The separation efficiency was calculated to be ~99.5%. It is worth noting that only the self-weight of the mixture was employed to drive the separation process, indicating the ease of operation and low energy consumption of the process.

## 4. Conclusions

Porous ceramic membranes prepared using the phase-inversion tape casting method possessed straight finger-like pores, presenting a small resistance to mass transfer. The ceramic membrane was hydrophilic in nature and could be converted to hydrophobic by grafting with FAS. FAS molecules were coupled on the alumina surface in both physical and chemical ways. The hydrophobic coating layer was stable in air and needs to be recovered periodically in practical applications. The hydrophilic membrane can be used to remove water from its mixture with oil, while the hydrophobic membrane can be applied to separate oil from its mixture with water. Due to the small mass transfer resistance of the membrane, the separation process can be driven solely by the weight of the mixture, thus it is highly energy efficient. In comparison with polymer and metal membranes, ceramic membranes exhibited better chemical and mechanical stability, which is promising for industrial and environmental applications requiring oil–water separation.

## Figures and Tables

**Figure 1 materials-14-00457-f001:**
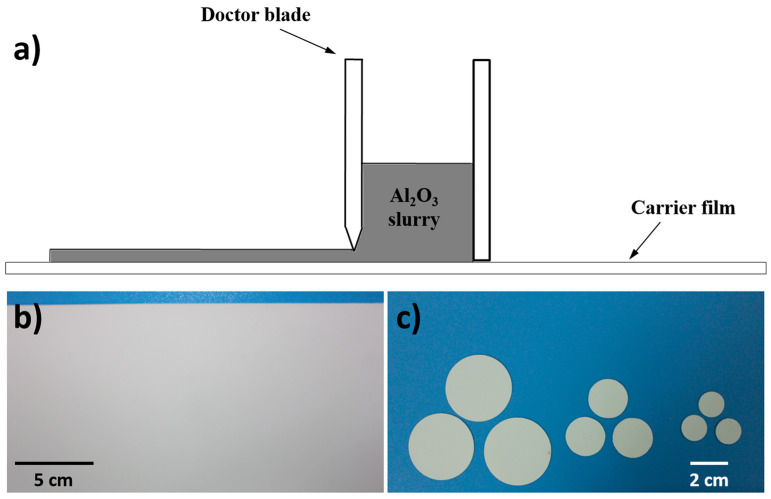
(**a**) Schematic illustration of the tape casting method, (**b**) alumina green tape after phase-inversion and drying, and (**c**) green samples after cutting.

**Figure 2 materials-14-00457-f002:**
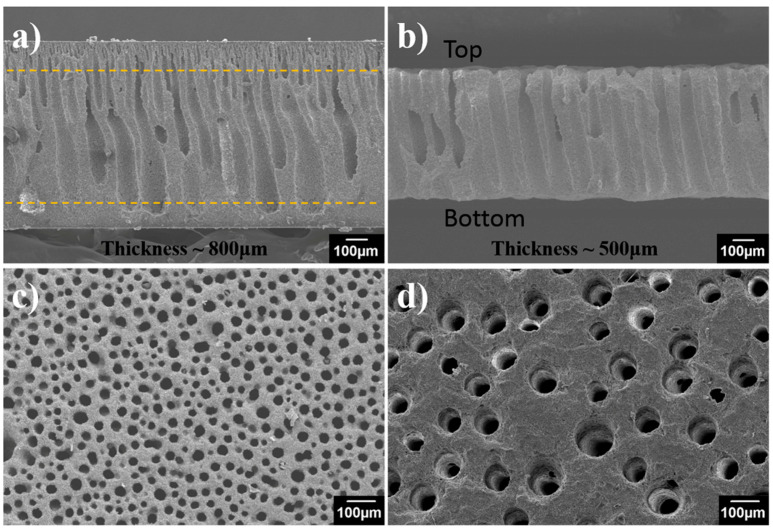
Scanning electron microscope (SEM) images of the as-prepared alumina membrane. (**a**) Cross-section before polishing, (**b**) cross-section after polishing, (**c**) top surface, and (**d**) bottom surface.

**Figure 3 materials-14-00457-f003:**
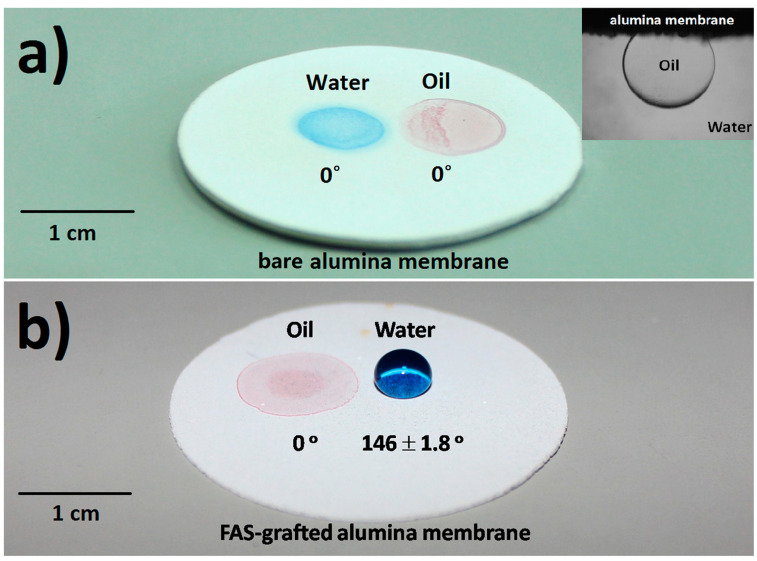
Wetting behavior of water and oil on alumina membranes (top surface). (**a**) Droplets of water (dyed blue, 50 μL) and oil (octane, dyed red, 50 μL) on a bare alumina membrane. (Inset) A photograph of an oil droplet (octane, 5 μL) underwater on a bare alumina membrane. (**b**) Droplets of water (dyed blue) and oil (octane, dyed red) on a fluoroalkylsilane (FAS)-grafted alumina membrane.

**Figure 4 materials-14-00457-f004:**
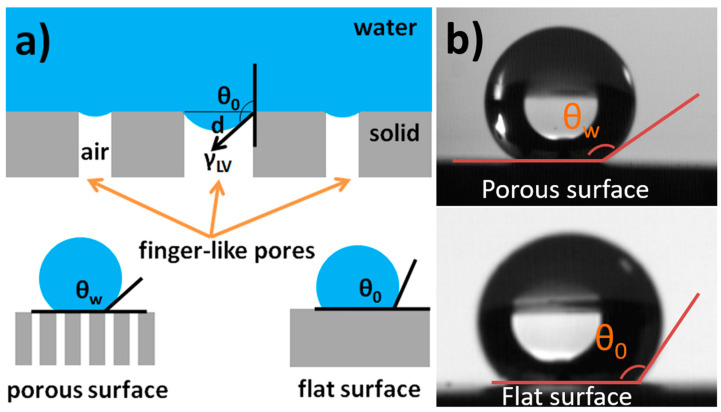
(**a**) Illustration of the calculation of theoretical water intrusion pressure of FAS-grafted membrane. *θ_w_* and *θ*_0_ are the apparent water contact angle on a porous hydrophobic surface and a flat hydrophobic surface, respectively. (**b**) Photographs of a water droplet (5 μL) on the as-prepared porous hydrophobic surface (polished, FAS-grafted) and flat hydrophobic surface (unpolished, FAS-grafted). *θ_w_* was ~146° and *θ*_0_ was ~131° in our case.

**Figure 5 materials-14-00457-f005:**
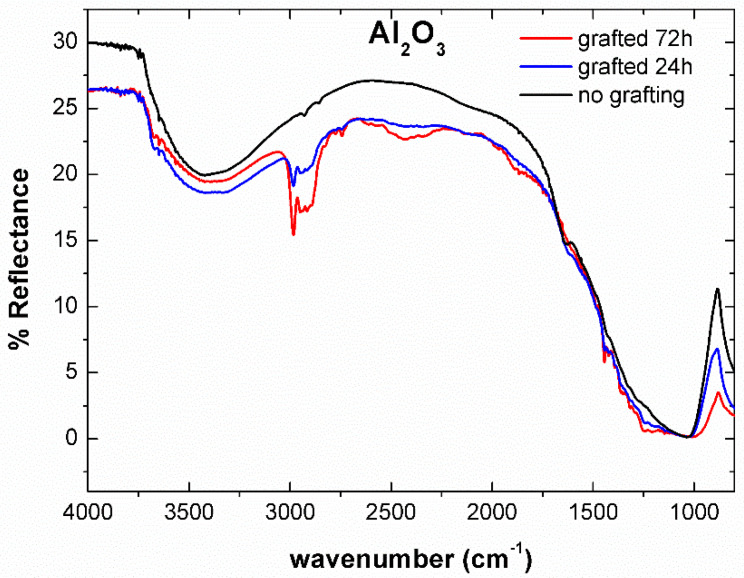
DRIFTS spectrum of the alumina powders before and after grafting with FAS.

**Figure 6 materials-14-00457-f006:**
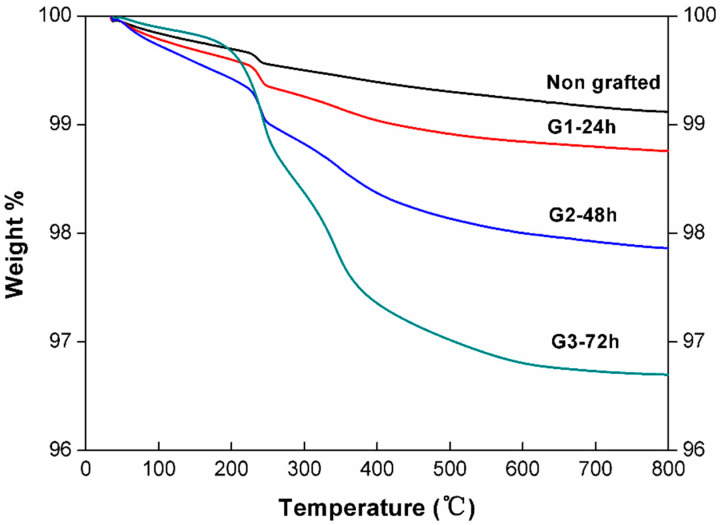
TG analysis of the alumina powder without grafting and the FAS-grafted alumina powder at different times. G1-24 h, G2-48 h, and G3-72 h, respectively.

**Figure 7 materials-14-00457-f007:**
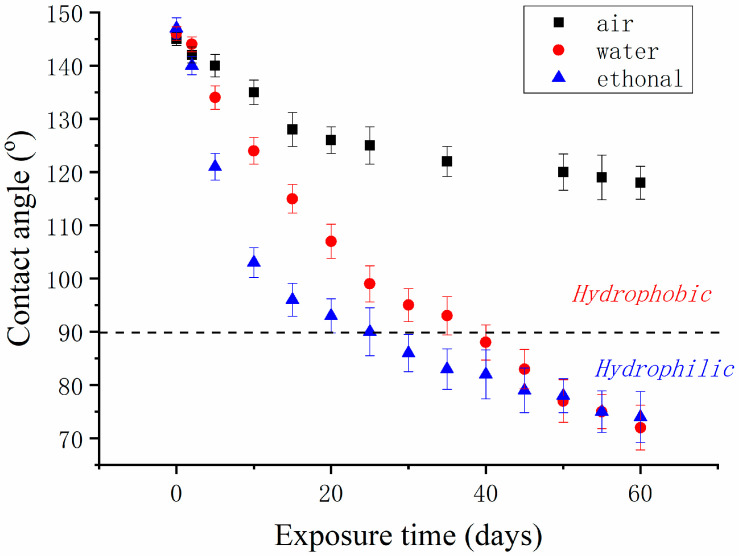
Change of water contact angle of FAS-grafted alumina membrane after exposure to air, water, and ethanol, respectively, as a function of exposure time (days).

**Figure 8 materials-14-00457-f008:**
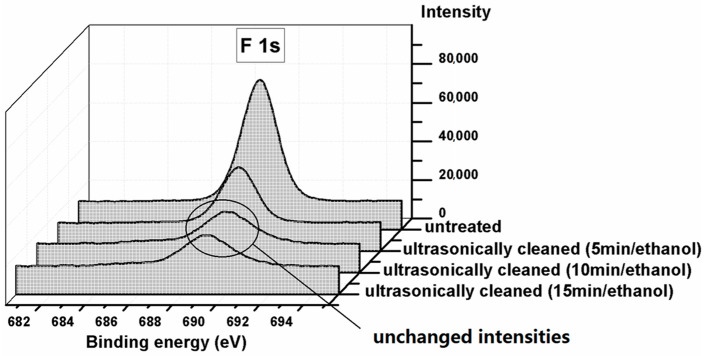
XPS spectra of F 1s on FAS-grafted alumina membrane before and after ultrasonic cleaning in ethanol.

**Figure 9 materials-14-00457-f009:**
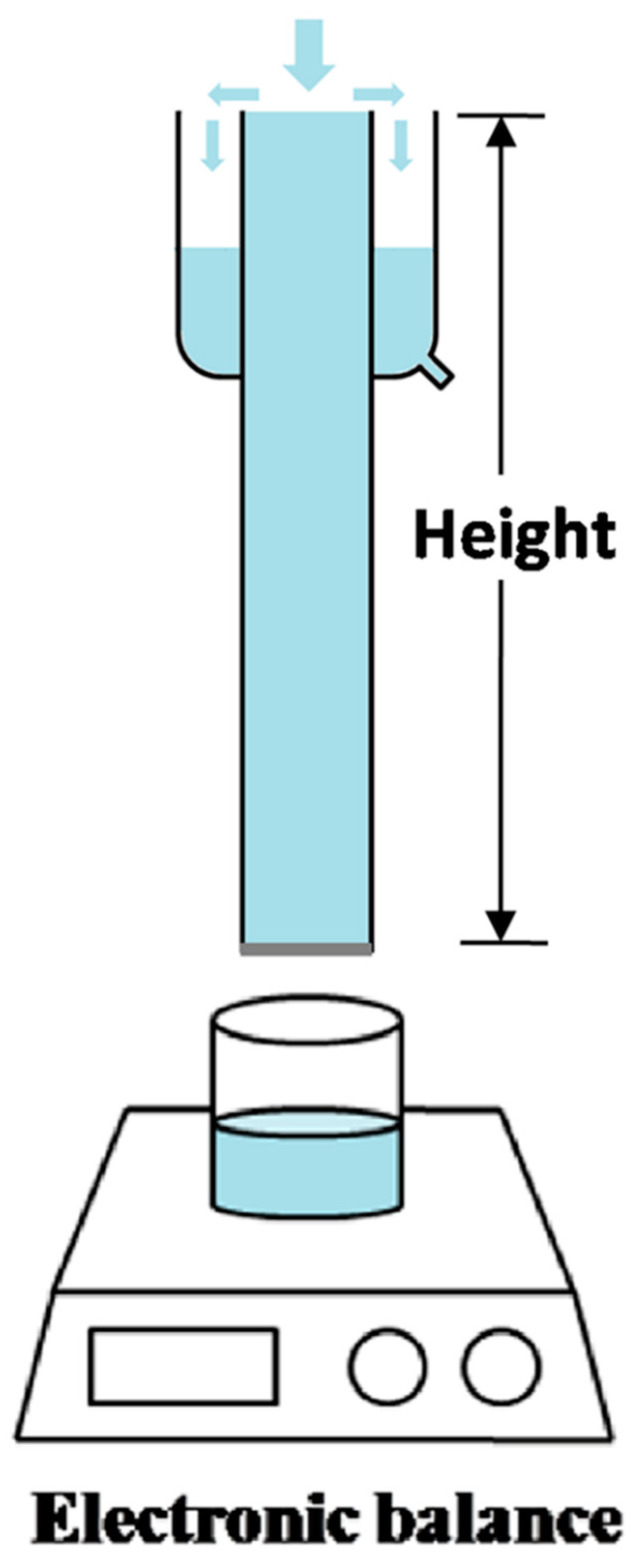
Schematic setup for the octane or water permeation tests.

**Figure 10 materials-14-00457-f010:**
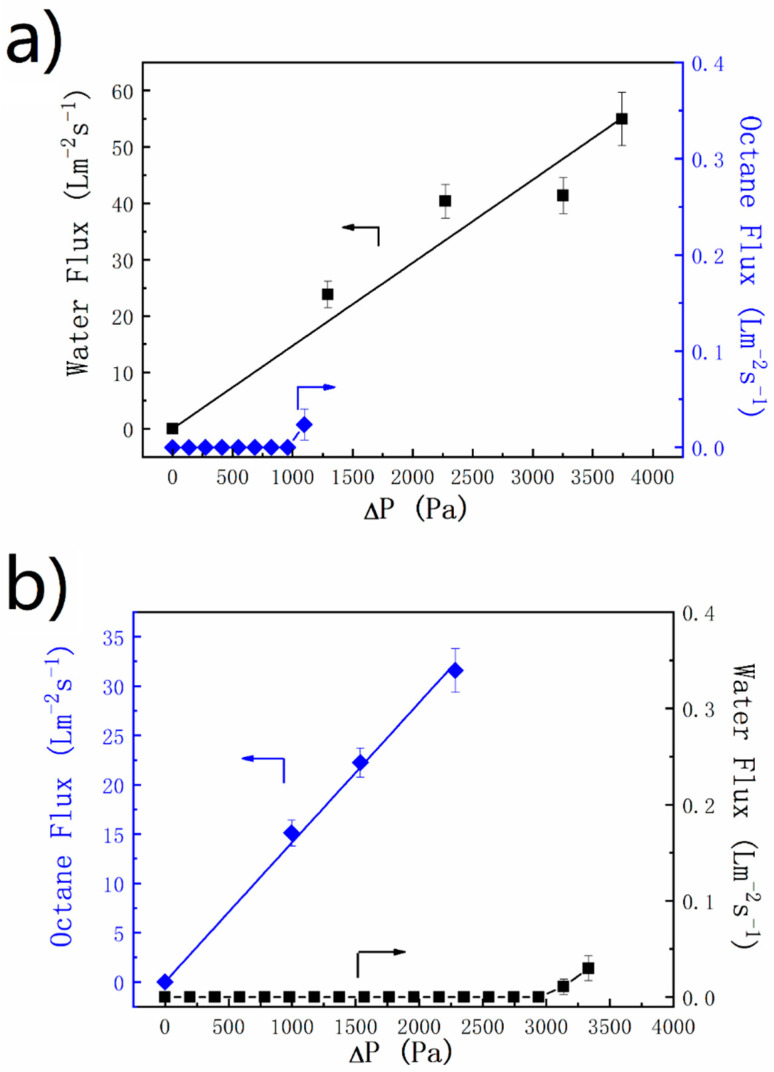
Water and octane flux of (**a**) a bare alumina membrane and (**b**) a FAS-grafted alumina membrane.

**Figure 11 materials-14-00457-f011:**
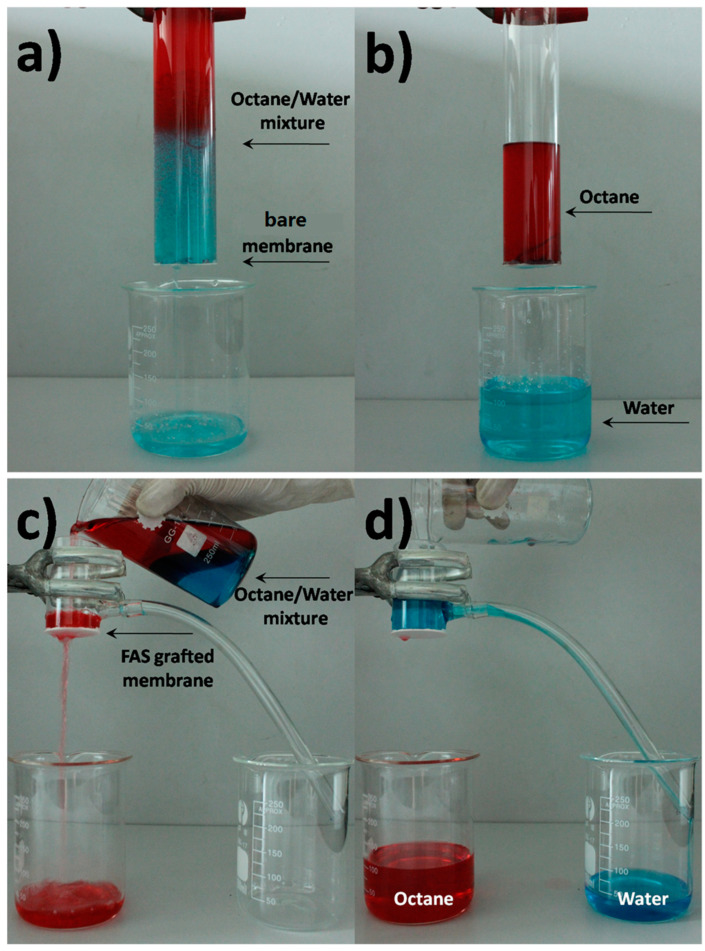
Oil/water separation equipment. (**a**) Octane/water mixture (30/70 *v*/*v*) above a porous alumina membrane and water was poured into the beaker. (**b**) Water selectively permeated through the bare membrane, while the octane was repelled and remained in the upper glass tube. (**c**) Separation of octane/water mixture (70/30 *v*/*v*) using the FAS-grafted membrane; octane easily passed through the membrane and flowed into a beaker. (**d**) Water was repelled by the FAS-grafted membrane and flowed into another beaker.

## Data Availability

Research data can be found at the journal of Materials or from the authors.

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
