# Peer review of "Gravity-Driven Separation of Oil/Water Mixture by Porous Ceramic Membranes with Desired Surface Wettability"

_materials, 2021, doi:10.3390/ma14020457_

Round 1

Reviewer 1 Report

The authors present an interesting and innovative way of research about the gravity-driven separation of oil/water mixture by porous ceramic membranes with desired surface wettability. In my opinion, this document does provide sufficient novelty in the area to be considered for publication. The authors presented research that is of great importance for the industry. This is a key topic for separation of oil and water mixture is becoming a huge worldwide challenge because of the still growing amount of industrial oil-containing wastewaters and of oil spill accidents. I believe the topic discussed in this article fully matches the magazine such as "Materials". The title of the article is satisfactory. The abstract covers pertinent points. The scientific quality of the article is good.

The presented data are reliable and useful. However, the paper needs a few improvements after which it can be published.

Personally, I think, the introduction is too general. It is necessary to strongly improve this part in order to give a complete scientific framework of the proposed research. In my opinion, the authors of the manuscript should necessarily refer to the newer publications from 2020-2021 in the "Introduction" section.  It seems to me that there are many curious and interesting works about porous Al2O3 membranes that are prepared through a phase inversion tape casting/sintering method that had been published in recent years, which could be referred to in the "Introduction" part of the manuscript.

According to the authors of the manuscript due to the excellent oil/water separation performance and good chemical stability, the porous ceramic membranes described in the article hold promise for practical applications. This begs the question of whether the membranes produced were comparable to commercially available membranes? Do the authors of the manuscript plan to conduct a market analysis on the basis of which they will be able to compare the obtained product with commercially available membranes? Is the current stage of research led by the authors are only experiments laboratory scale?

Author Response

Personally, I think, the introduction is too general. It is necessary to strongly improve this part in order to give a complete scientific framework of the proposed research. In my opinion, the authors of the manuscript should necessarily refer to the newer publications from 2020-2021 in the "Introduction" section.  It seems to me that there are many curious and interesting works about porous Al2O3 membranes that are prepared through a phase inversion tape casting/sintering method that had been published in recent years, which could be referred to in the "Introduction" part of the manuscript.

Response: The introduction part is re-organized according to the reviewer’s suggestion. Specifically, the references have been updated to the most recent works. Studies referring to the phase inversion tape casting methods for porous ceramic membranes have been cited and discussed in the revised manuscript, in order to highlight the value and complete the scientific framework of our conducted research.

According to the authors of the manuscript due to the excellent oil/water separation performance and good chemical stability, the porous ceramic membranes described in the article hold promise for practical applications. This begs the question of whether the membranes produced were comparable to commercially available membranes? Do the authors of the manuscript plan to conduct a market analysis on the basis of which they will be able to compare the obtained product with commercially available membranes? Is the current stage of research led by the authors are only experiments laboratory scale?

Response: We totally agree with the reviewer’s comment about the statements of “the porous ceramic membranes described in the article hold promise for practical applications” in the previous manuscript. We have used “display potential” instead of “hold promise” in the revised manuscript. It is also important noting that our research is lab-scale based work and need to be further studied before being considered for practical use.

Reviewer 2 Report

The authors have not mentioned, how the pore size of the membranes were measured.

Some of the important references ( for example,Materials 2020, 13, 3354; Membranes,  2019, 9, 128)

should be quoted after improving the introduction for the sake of readers to understand the importance of this field (such as ceramic membranes).

Author Response

all the references has been added.

Reviewer 3 Report

This manuscript reports the preparation of porous Al2O3 membranes through a phase-inversion tape casting/sintering method, to be used for oil/water separation. The authors gave a good introduction for the need for such membranes, the method of preparing membranes, experimental methods, results, and discussion. I would appreciate the work of the authors to detail the work and report the advantages of adding PAS to the alumina membranes. I would recommend the acceptance of this manuscript after minor revisions as mentioned below:

  1. Methods – How did the authors choose the PES and Al2O3 amount (in %) to fabricate the planar alumina membranes? Which is the polymer binder that was removed here? Also, can the authors add reference to such preparation method?
  2. How did the authors confirm the grafting of FAS to the alumina membranes?
  3. The authors could add the error bars for data in Figure 6 and 9 to increase the reliability of the data.
  4. Spelling corrections and English check is to be done

Author Response

1. Methods – How did the authors choose the PES and Al2O3 amount (in %) to fabricate the planar alumina membranes? Which is the polymer binder that was removed here? Also, can the authors add reference to such preparation method?

Response: Thanks for the reviewer’s comment. Multiple changes have been made in the Introduction and Method parts of the revised manuscript. The phase inversion method to prepare a porous ceramic membrane is detailed, including the pore-forming mechanism and the most recent developments. Moreover, the PES and alumina weight ratio were optimized in early works and have been cited in the revised manuscript. The PESf, NMP, and PVP were used as a polymer binder, a solvent, and an additive, respectively.

2. How did the authors confirm the grafting of FAS to the alumina membranes?

Response: We have added the explanation “The grafting process can be performed by a reaction between -OH surface groups of the ceramics and ethoxy groups (-O-Et) presented in organosilane compounds.” In the revised manuscript. And the IR spectrum of the alumina powders before and after grafting FAS is added in the revised manuscript. As shown in Figure 5. the FAS grafting was confirmed by the intensity decrease of surface hydroxyl-related absorption, as well as the arise of C-F and C-H bonds. Besides, the TG results and XPS F1s peaks in Figures 6 and 8 also verified the existence of FAS molecules on the grafted alumina surface.

3. The authors could add the error bars for data in Figure 6 and 9 to increase the reliability of the data.

Response: Thanks for the comments, it is very important to summarize the data in a more reliable way. We have added the error bars for the date in these Figures. Moreover, we used pressure difference instead of liquid height for the x-axis in Figure 7, for a clear understanding of the relationship between flux and driven force.

4.Spelling corrections and English check is to be done

Response: Spelling issues are carefully checked and revised, which have been highlighted in the present manuscript.
